# Impact of serum phosphate levels during CRRT on extubation failure and hospital mortality in mechanically ventilated ICU patients—A study based on the MIMIC-IV database

Yucheng Li[1☯], Chuanyan Zhao[2☯], Xingjie Ma[1], Yunlong Pei[3], Weili Liu[1], Liang Gao[4*]

**1** Laboratory of Intensive Care Unit, Department of Intensive Care Unit, The Affiliated Hospital of Yangzhou University, Yangzhou University, Yangzhou, China, **2** Department of Nephrology, Northern Jiangsu People's Hospital, Yangzhou, China, **3** School of Biochemistry and Immunology, Trinity Biomedical Sciences Institute, Trinity College Dublin, Dublin, Ireland, **4** Department of Neurosurgery, Shanghai Tenth People's Hospital, Tongji University School of Medicine, Shanghai, China

☯ These authors contributed equally to this work.
* lianggaohhh@126.com

## Abstract

### Background

Electrolyte imbalances, particularly phosphate depletion, are prevalent yet often underestimated complications in the Intensive Care Unit (ICU), notably among patients undergoing Continuous Renal Replacement Therapy (CRRT). Therefore, this study aims to examine the impact of serum phosphate levels during CRRT on the incidence of extubation failure and hospital mortality in mechanically ventilated patients.

### Methods

Patients subjected to both CRRT and mechanical ventilation were extracted from the MIMIC-IV database. Cox regression analysis was employed to identify the potential risk factors for the extubation failure and hospital mortality rates. Patients were categorized into three groups based on their minimum serum phosphate level (Phosphate_min) during CRRT. Kaplan-Meier survival analysis and Receiver Operating Characteristic (ROC) curves were employed to assess differences in the primary outcomes among these groups. Additionally, a restricted cubic spline curve was utilized to explore potential nonlinear relationships between Phosphate_min and the primary outcomes.

### Results

The analysis included 816 ICU patients undergoing CRRT and mechanical ventilation. Cox regression analysis identified Phosphate_min as a significant risk factor

**Data availability statement:** All relevant data can be obtained from the MIMIC IV database (https://mimic.mit.edu/docs/iv/), and the data extracted from the MIMIC IV database is included in the Supporting Information file.

**Funding:** Northern Jiangsu People's Hospital under Grant (SBQN23009). The funder (Northern Jiangsu People's Hospital) and the first author contributed equally to this work, including study design, data analysis, and manuscript preparation. The funder had no role in the final decision to publish.

**Competing interests:** The authors have declared that no competing interests exist.

for both extubation failure (HR 1.29; 95% CI 1.22–1.36, $p < 0.001$) and hospital mortality (HR 1.48; 95% CI 1.37–1.60, $p < 0.001$). ROC curve analysis indicated that a Phosphate_min > 4.5 mg/dL was a moderate predictor for both extubation failure and hospital mortality. Kaplan-Meier analysis revealed significantly higher risks of the primary outcomes in the group with Phosphate_min > 4.5 mg/dL compared to the lower phosphate groups (log-rank $p < 0.001$). Additionally, restricted cubic spline analysis showed a J-shaped relationship between Phosphate_min and both primary outcomes, with nadirs at approximately 1.60 mg/dL for extubation failure and 1.98 mg/dL for hospital mortality.

## Conclusion

Phosphate_min emerges as an independent risk factor for both extubation failure and hospital mortality. Maintaining serum phosphate levels within a therapeutic range may potentially mitigate these risks.

## 1. Introduction

Continuous Renal Replacement Therapy (CRRT) is widely embraced as a treatment modality for critically ill patients. According to a multicenter cross-sectional study in 2015, approximately 12.0% to 15.1% of Intensive Care Unit (ICU) inpatients underwent CRRT [1], primarily due to its favorable tolerability among individuals with hemodynamic instability [2]. However, it is imperative to monitor the loss of electrolytes, including phosphate, during CRRT to avert potential imbalances. Previous investigations have revealed varying degrees of phosphate loss during CRRT, with significant losses reported in up to 78% of cases [3–6].

Phosphate, an essential intracellular anion in the human body, plays a vital role in numerous physiological functions. It is integral to bone synthesis, the formation of biomembrane phospholipids, nucleic acids metabolism, regulation of enzyme activity, the creation of buffer systems, and maintaining the body's acid-base balance. Additionally, it serves as a critical component of adenosine triphosphate (ATP), the primary energy carrier in cells. Imbalances in phosphate levels can lead to various clinical conditions, including tissue hypoxia, systemic muscle weakness, reduced peripheral vascular resistance, impaired myocardial contraction, arrhythmias, metabolic encephalopathy, and potentially mortality [7–11].

In critical care settings, particularly among patients requiring mechanical ventilation, deviations from optimal phosphate levels can have serious consequences. Low phosphate levels may induce respiratory muscle weakness, prolonging the need for mechanical ventilation, increasing the risk of reintubation, and extending ICU stay and hospitalization time [12–14]. Conversely, high phosphate levels have been linked to increased hospital mortality rates [15]. While previous studies have extensively examined the effects of phosphate levels on mechanical ventilation duration and mortality in non-CRRT patients [15], as well as the dynamics of phosphate levels during CRRT [16], the specific association between serum phosphate levels during CRRT

and critical outcomes such as extubation failure and hospital mortality in mechanically ventilated patients remains largely unknown.

Given that CRRT can induce substantial shifts in serum phosphate levels, we hypothesized that these alterations might impact extubation outcomes and hospital mortality among mechanically ventilated patients. To investigate this hypothesis, we conducted a retrospective analysis using data from the Medical Information Mart for Intensive Care IV (MIMIC-IV) database. Our objective is to assess the potential relationship between serum phosphate levels during CRRT and critical outcomes in mechanically ventilated ICU patients.

## 2. Materials and methods

### 2.1. Study design and population

The study relied on the MIMIC-IV electronic database (version 2.2), a collaborative effort between the Massachusetts Institute of Technology (MIT) and Beth Israel Deaconess Medical Center (BIDMC). This extensive database contains comprehensive data on patients receiving inpatient care at BIDMC from 2008 to 2019. Approval from the Institutional Review Board (IRB) of BIDMC waived the need for informed consent and authorized the sharing of research resources due to the thorough de-identification process applied to all data [17]. Before data extraction, Yucheng Li, the author, met all necessary requirements for accessing the database. The inclusion criteria were established as follows: patients admitted to the hospital and ICU for the first time, aged 18 years or older, requiring both CRRT and mechanical ventilation. Furthermore, patients needed to have a minimum duration of mechanical ventilation and ICU stay of at least 48 hours (refer to Fig 1).

### 2.2. Data collection and variable definition

The pgAdmin PostgreSQL tool (version 6.1) was utilized for data extraction from the MIMIC-IV database, encompassing the collection of the following variables: age, gender, Body Mass Index (BMI), Simplified Acute Physiology Score II (SAPS II), Sequential Organ Failure Assessment (SOFA) score on the first day of ICU admission (SOFA score_first day), CRRT duration, initial phosphate level after ICU admission (Phosphate_first), maximum serum phosphate level during CRRT (Phosphate_max), minimum serum phosphate level during CRRT (Phosphate_min), mean serum phosphate level during CRRT (Phosphate_mean), primary diagnosis upon admission, length of ICU stay (ICU LOS), length of hospital stay (Hospital LOS), mechanical ventilation duration and frequency of intubation.

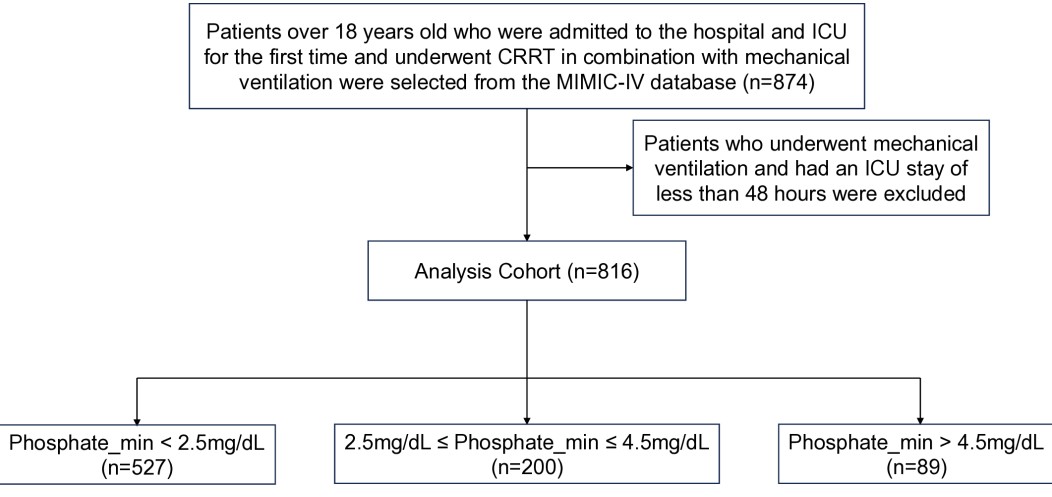

**Fig 1. Flow chart of patient selection.**

In this study, extubation failure is defined as the need for unplanned reintubation within 48 hours following a planned extubation attempt, or failure to maintain unassisted spontaneous breathing for ≥48 hours [18]. The reference range for phosphate levels was set between 2.5 to 4.5 mg/dL. Consequently, patients were categorized into three groups according to their phosphate levels: those below 2.5 mg/dL, those within the range of 2.5 to 4.5 mg/dL, and those exceeding 4.5 mg/dL.

To address potential bias, variables with missing values exceeding 20% were excluded. However, for variables with less than 20% missing data, imputation was performed using mean values only (with BMI being the sole variable with missing values). Our primary outcomes included extubation failure and hospital mortality, while secondary outcomes comprised ICU length of stay.

## 2.3. Statistical analyses

The data analysis employed R software (Version 4.2.2) and SPSS statistical software (Version 26.0). Normal distributions of variables were assessed using Shapiro-Wilk tests. The statistical analyses encompassed t-tests or analysis of variance (ANOVA). Continuous variables were reported as mean ± standard deviation or median (interquartile range, IQR). Categorical variables were presented as numbers (proportions) and analyzed using chi-square tests, corrected chi-square tests, or Fisher's exact test as appropriate.

Univariable Cox regression analysis was conducted to explore the association between independent variables and hospital mortality and extubation outcome. Variables demonstrating clinical significance and a significance level of $p < 0.15$ were selected for inclusion in the multivariable Cox regression model. Kaplan-Meier survival analysis was used to gauge the incidence rate of major outcome events across various stratified groups based on the independent variable with the highest risk ratio, with differences assessed through log-rank tests. Receiver Operating Characteristic (ROC) curve analysis was employed to assess the predictive ability, sensitivity, and specificity of the independent variable with the highest risk ratio for predicting the primary outcome. In addition, restricted cubic spline analysis was utilized to capture any dose-effect relationship between the independent variable with the highest risk ratio and the risk of primary outcome occurrence. A two-sided P-value less than 0.05 was deemed statistically significant for all analyses.

## 3. Results

### 3.1. Baseline characteristics and clinical course

In this study, 816 patients were enrolled, and their selection criteria are outlined in Fig 1. The demographic and clinical profiles of patients who experienced successful extubation versus those who did not, are delineated in Table 1. Additionally, Table 2 provides a comparison between patients who passed away outside the hospital and those who expired within hospital premises. While age did not exhibit significant discrepancies between the successful and unsuccessful extubation groups, a notable observation emerged regarding patient survival. Specifically, survivors tended to be younger than individuals who succumbed within the hospital (mean ages: 60.43 ± 15.36 vs. 63.81 ± 14.90, p = 0.002). Other parameters such as gender, BMI, SAPS II score, SOFA score on the first day, first-day phosphate levels, and duration of mechanical ventilation did not manifest significant differences among the groups. However, distinct differences emerged concerning phosphate levels. Patients who underwent successful extubation or survived during hospitalization showcased lower maximum, minimum, and mean phosphate levels ($p < 0.05$, $p < 0.001$, and $p < 0.001$, respectively) compared to those in the failed extubation group. Conversely, the frequency of intubation was notably higher among patients who successfully underwent extubation ($p < 0.001$). Furthermore, both ICU and hospital length of stay (LOS) were significantly prolonged for patients who experienced failed extubations or passed away within the hospital ($p < 0.001$ for both).

### 3.2. Univariable and multiviable Cox regression analysis of the primary outcomes

To gauge the relative influence of our selected independent variables on the outcomes of extubation failure and hospital mortality, we initially conducted a univariable Cox regression analysis to scrutinize each independent variable (refer to

**Table 1. Demographic and clinical characteristics of patients with successful and failed extubation.**

| Variables | ALL | Successful Extubation | Failde Extubation | *p* Value |
|---|---|---|---|---|
| | n = 816 | n = 440 | n = 376 | |
| Age (years) | 62.07 (15.22) | 61.41 (15.22) | 62.86 (15.21) | 0.175 |
| Gender (Male (%)) | 507 (62.1) | 271 (61.6) | 236 (62.8) | 0.785 |
| BMI (kg/m2) | 31.66 (8.56) | 31.55 (8.27) | 31.78 (8.91) | 0.703 |
| SAPSIIscore | 55.51 (15.21) | 54.88 (14.86) | 56.26 (15.60) | 0.196 |
| SOFA score _first day | 12.96 (3.81) | 12.83 (3.66) | 13.11 (3.97) | 0.296 |
| CRRT time (hours) | 98.61 [47.47, 188.57] | 118.48 [64.27, 215.48] | 70.03 [32.93, 151.21] | <0.001 |
| Phosphate_first (mg/dL) | 5.00 [3.70, 6.53] | 5.00 [3.70, 6.60] | 4.95 [3.70, 6.50] | 0.848 |
| Phosphate_max (mg/dL) | 5.70 [4.40, 7.20] | 5.50 [4.30, 7.10] | 5.80 [4.50, 7.30] | 0.049 |
| Phosphate_min (mg/dL) | 2.00 [1.50, 3.10] | 1.90 [1.40, 2.50] | 2.30 [1.60, 3.82] | <0.001 |
| Phosphate_mean (mg/dL) | 3.38 [2.70, 4.40] | 3.24 [2.65, 4.12] | 3.58 [2.79, 4.91] | <0.001 |
| Duration of mechanical ventilation (hours) | 137.38 [61.04, 254.92] | 141.50 [58.19, 254.58] | 131.82 [64.72, 257.53] | 0.615 |
| Intubation frequency | 1.00 [1.00, 3.00] | 2.00 [1.00, 3.00] | 1.00 [1.00, 2.00] | <0.001 |
| ICU LOS (days) | 11.39 [6.73, 19.47] | 14.42 [9.34, 24.10] | 8.12 [4.30, 13.81] | <0.001 |
| Hospital LOS (days) | 19.68 [10.60, 32.65] | 26.94 [18.52, 42.49] | 11.14 [4.93, 19.54] | <0.001 |
| First diagnosis during hospitalization, n (%) | | | | |
| Cardiovascular disease | 206 (25.25) | 123(27.95) | 83(22.07) | 0.054 |
| Digestive system disease | 153 (18.75) | 88 (2.00) | 65(17.29) | 0.322 |
| Hematological disease | 17 (2.08) | 9 (2.05) | 8 (2.13) | 0.935 |
| Kidney disease | 23 (2.82) | 15 (3.41) | 8 (2.13) | 0.270 |
| Nervous system disease | 15 (1.84) | 3(0.68) | 12 (3.19) | 0.009 |
| Postoperative complication | 53 (6.50) | 23 (5.23) | 30 (7.98) | 0.112 |
| Rheumatic immune disease | 6 (0.73) | 1(0.23) | 5 (1.33) | 0.100 |
| Respiratory disease | 46 (5.64) | 19(4.32) | 27 (7.18) | 0.077 |
| Sepsis | 226 (27.70) | 122 (27.73) | 104(27.66) | 0.983 |
| Toxicosis | 19 (2.33) | 11 (2.50) | 8 (2.13) | 0.725 |
| Trauma | 16 (1.96) | 10 (2.27) | 6 (1.60) | 0.487 |
| Tumor | 7 (0.86) | 5 (1.14) | 2 (0.53) | 0.461 |
| Other | 29 (3.55) | 11(2.50) | 18(4.79) | 0.079 |

BMI: body mass index; SAPSII score: simplified acute physiology score II; SOFA score _first day: Sequential Organ Failure Assessment score on the first day of ICU admission; Phosphate_first: the first measurement of serum phosphate level after admission to the ICU; Phosphate_max: maximum serum serum phosphate level during CRRT; Phosphate_min: minimum serum phosphate level during CRRT; Phosphate_mean: mean value of serum phosphate level during CRRT; ICU LOS: length of stay in ICU; Hospital LOS: length of stay in hospital.

Table 3). Variables demonstrating clinical significance and attaining a significance level of P < 0.15 were deemed eligible for inclusion in the multivariable Cox regression model. Notably, due to collinearity concerns, Phosphate_mean was excluded from this analysis. The findings unveiled that Phosphate_min emerged as a significant predictor for both the likelihood of extubation failure (HR 1.29; 95% CI 1.22–1.36, p < 0.001) and hospital mortality (HR 1.48; 95% CI 1.37–1.60, p < 0.001) (Figs 2 and 3).

### 3.3. Serum phosphate levels in relation to extubation failure and hospital mortality

The Kaplan-Meier survival analysis curves depicted in Figs 4 and 5 vividly illustrate the influence of serum phosphate levels on both extubation failure and hospital mortality. Notably, patients with Phosphate_min levels below 2.5 mg/dL exhibited significantly lower incidences of failed extubation and hospital mortality compared to individuals falling into the

**Table 2. Demographic and clinical characteristics of patients with hospital and non-hospital deaths.**

| Variables | ALL | Non-Hospital death | Hospital death | p Value |
|---|---|---|---|---|
| | n = 816 | n = 419 | n = 397 | |
| Age (years) | 62.07 (15.22) | 60.43 (15.36) | 63.81 (14.90) | 0.002 |
| Gender (Male (%)) | 507 (62.1) | 257 (61.3) | 250 (63.0) | 0.682 |
| BMI (kg/m2) | 31.66 (8.56) | 31.49 (8.35) | 31.84 (8.79) | 0.555 |
| SAPSIIscore | 55.51 (15.21) | 55.01 (14.77) | 56.05 (15.66) | 0.331 |
| SOFA score _ first day | 12.96 (3.81) | 12.81 (3.65) | 13.12 (3.96) | 0.258 |
| CRRT time (hours) | 98.61 [47.47, 188.57] | 111.20 [58.31, 203.14] | 86.12 [37.68, 171.65] | 0.001 |
| Phosphate_first (mg/dL) | 5.00 [3.70, 6.53] | 5.00 [3.80, 6.60] | 5.00 [3.70, 6.40] | 0.940 |
| Phosphate_max (mg/dL) | 5.70 [4.40, 7.20] | 5.50 [4.25, 7.00] | 5.90 [4.60, 7.30] | 0.007 |
| Phosphate_min (mg/dL) | 2.00 [1.50, 3.10] | 1.90 [1.50, 2.60] | 2.30 [1.50, 3.70] | <0.001 |
| Phosphate_mean (mg/dL) | 3.38 [2.70, 4.40] | 3.23 [2.64, 4.16] | 3.54 [2.82, 4.77] | <0.001 |
| Duration of mechanical ventilation (hours) | 137.38 [61.04, 254.92] | 142.62 [59.26, 251.12] | 127.52 [63.53, 261.00] | 0.823 |
| Intubation frequency | 1.00 [1.00, 3.00] | 2.00 [1.00, 3.00] | 1.00 [1.00, 2.00] | <0.001 |
| ICU LOS (days) | 11.39 [6.73, 19.47] | 13.28 [8.88, 22.73] | 9.52 [4.85, 16.51] | <0.001 |
| Hospital LOS (days) | 19.68 [10.60, 32.65] | 26.98 [18.93, 41.92] | 11.22 [5.29, 20.46] | <0.001 |
| First diagnosis during hospitalization, n (%) | | | | |
| Cardiovascular disease | 206 (25.25) | 109 (26.01) | 97 (24.43) | 0.603 |
| Digestive system disease | 153 (18.75) | 79 (18.85) | 74 (18.64) | 0.937 |
| Hematological disease | 17 (2.08) | 9 (2.15) | 8 (2.02) | 0.895 |
| Kidney disease | 23 (2.82) | 18 (4.30) | 5 (1.26) | 0.009 |
| Nervous system disease | 15 (1.84%) | 3 (0.72) | 12 (3.02) | 0.018 |
| Postoperative complication | 53 (6.50%) | 25 (6.00) | 28 (7.05) | 0.529 |
| Rheumatic immune disease | 6(0.73) | 2 (0.48) | 4 (1.01) | 0.440 |
| Respiratory disease | 46 (5.64) | 18 (4.30) | 28 (7.05) | 0.088 |
| Sepsis | 226 (27.70) | 113 (26.97) | 113(28.46) | 0.634 |
| Toxicosis | 19 (2.33) | 15 (3.58%) | 4(1.01%) | 0.018 |
| Trauma | 16 (1.96) | 10 (2.39) | 6 (1.51) | 0.367 |
| Tumor | 7 (0.86) | 6 (1.43) | 1 (0.25) | 0.124 |
| Other | 29 (3.55) | 13 (3.10) | 16 (4.03) | 0.474 |

BMI: body mass index; SAPSII score: simplified acute physiology score II; SOFA score _first day: Sequential Organ Failure Assessment score on the first day of ICU admission; Phosphate_first: the first measurement of serum phosphate level after admission to the ICU; Phosphate_max: maximum serum phosphate level during CRRT; Phosphate_min: minimum serum phosphate level during CRRT; Phosphate_mean: mean value of serum phosphate level during CRRT; ICU LOS: length of stay in ICU; Hospital LOS: length of stay in hospital.

other two groups (log-rank p < 0.001). Conversely, patients with Phosphate_min levels exceeding 4.5 mg/dL demonstrated notably elevated rates of these adverse outcomes (log-rank p < 0.001).

The ROC curve served as a tool to assess the predictive efficacy of Phosphate_min regarding both extubation failure and hospital mortality. In predicting extubation failure, Phosphate_min levels above 4.5 mg/dL demonstrated a moderate predictive capacity, yielding an AUC of 0.806 (95% CI: 0.708, 0.904). The optimal cut-off point was determined as 5.250, with a specificity of 73.7% and sensitivity of 77.1%. In the context of predicting hospital mortality, Phosphate_min exhibited an AUC of 0.775 (95% CI: 0.676, 0.873). The ideal cut-off point was identified as 5.350, accompanied by a specificity of 80.0% and a sensitivity of 72.5% (Fig 6).

Restricted cubic splines were employed to flexibly model and visualize the relationship between predicted Phosphate_min levels and the primary outcomes, as depicted in Fig 7. The plot revealed a J-shaped relationship between predicted Phosphate_min and extubation failure. Notably, there was a significant reduction in risk within the lower range (< 1.60 mg/

**Table 3. Associated clinical factors for failure rate of extubation and hospital mortality by univariate Cox regression analysis.**

| Variables | Failure rate of extubation | | | Hospital mortality | | |
|---|---|---|---|---|---|---|
| | HR | 95% CI | *p* Value | HR | 95% CI | *p* Value |
| Gender | 0.99 | 0.89-1.36 | 0.376 | 0.96 | 0.78-1.17 | 0.671 |
| Age | 1.01 | 1.00-1.01 | 0.036 | 1.01 | 1.00-1.02 | <0.001 |
| BMI | 1.00 | 0.99-1.01 | 0.837 | 1.00 | 0.99-1.02 | 0.510 |
| SAPSIIscore | 1.01 | 1.00-1.01 | 0.041 | 1.01 | 1.00-1.01 | 0.012 |
| SOFA score_first day | 1.00 | 0.98-1.03 | 0.825 | 1.02 | 1.00-1.05 | 0.100 |
| CRRT time | 0.99 | 0.99-1.00 | <0.001 | 1.00 | 1.00-1.00 | <0.001 |
| Phosphate_first | 1.07 | 1.03-1.12 | 0.001 | 1.04 | 0.99-1.08 | 0.119 |
| Phosphate_max | 0.98 | 0.94-1.03 | 0.474 | 1.04 | 0.99-1.07 | 0.098 |
| Phosphate_min | 1.44 | 1.38-1.51 | <0.001 | 1.4383 | 1.36-1.52 | <0.001 |
| Intubation frequency | 0.68 | 0.63-0.73 | <0.001 | 0.8080 | 0.76-0.86 | <0.001 |

HR, Hazard Ratio; CI, confidence interval.

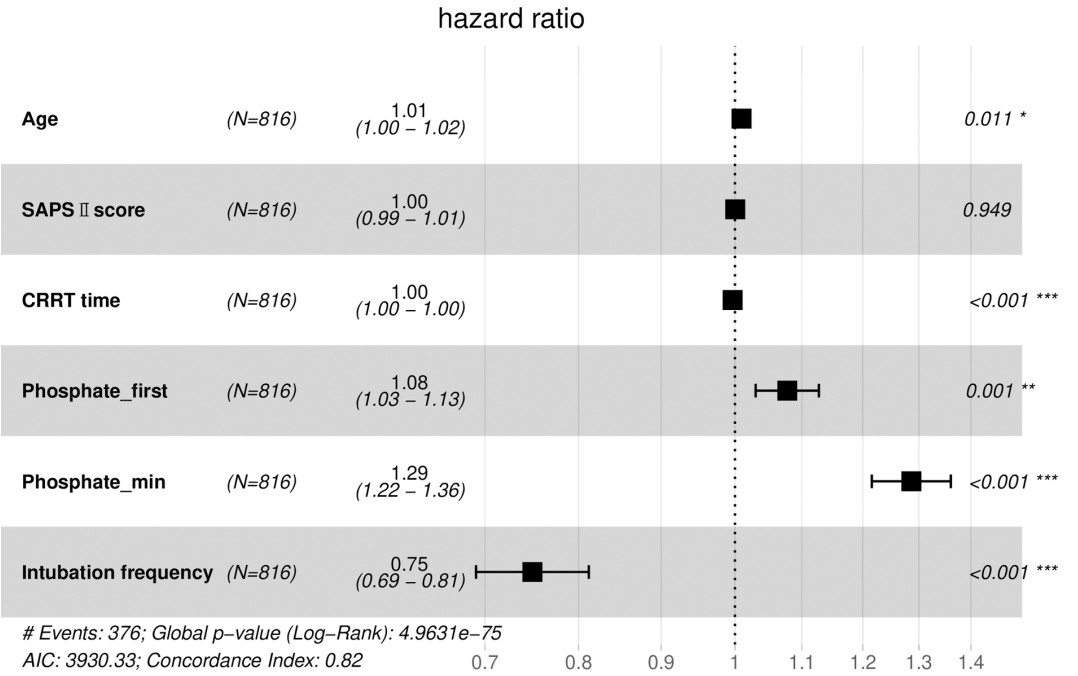

**Fig 2. Forest plot displaying the results of COX regression analysis of the extubation failure.**

dL), followed by an increase (p<0.001 for overall significance, p=0.048 for nonlinearity). Similarly, the risk of hospital mortality remained relatively stable until approximately a predicted Phosphate_min level of 1.98 mg/dL, after which it escalated rapidly (p<0.001 for overall significance, p=0.006 for nonlinearity).

## 4. Discussion

This study aimed to explore the impact of serum phosphate levels during CRRT on extubation failure and hospital mortality among mechanically ventilated patients. Our findings unveiled that Phosphate_min during CRRT treatment held

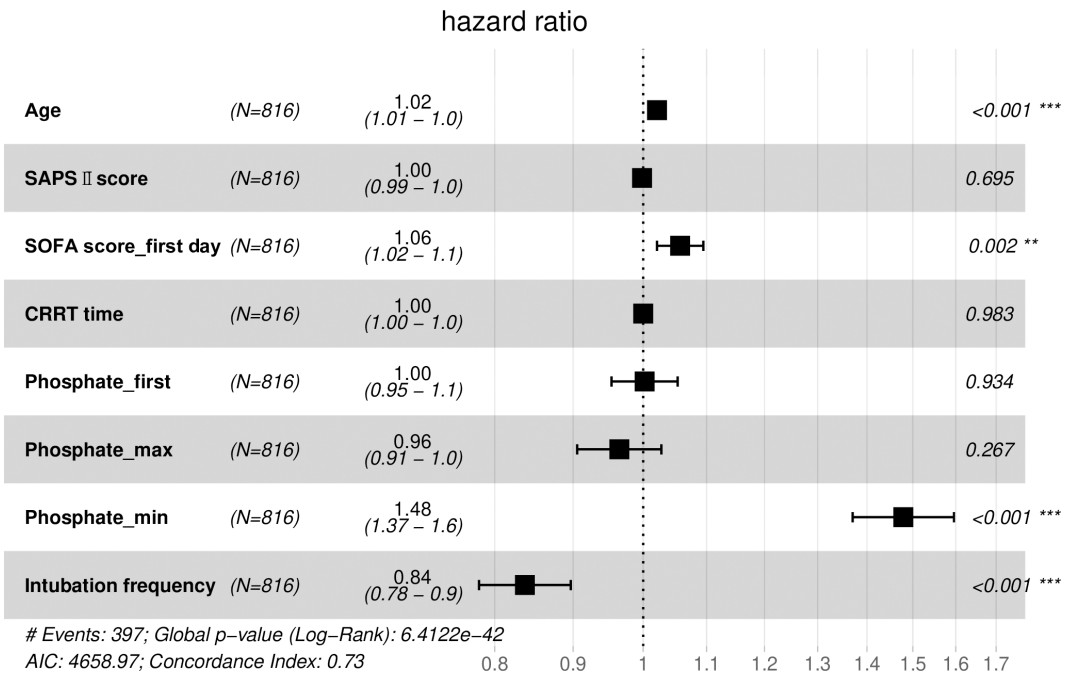

**Fig 3. Forest plot displaying the results of COX regression analysis of the hospital mortality.**

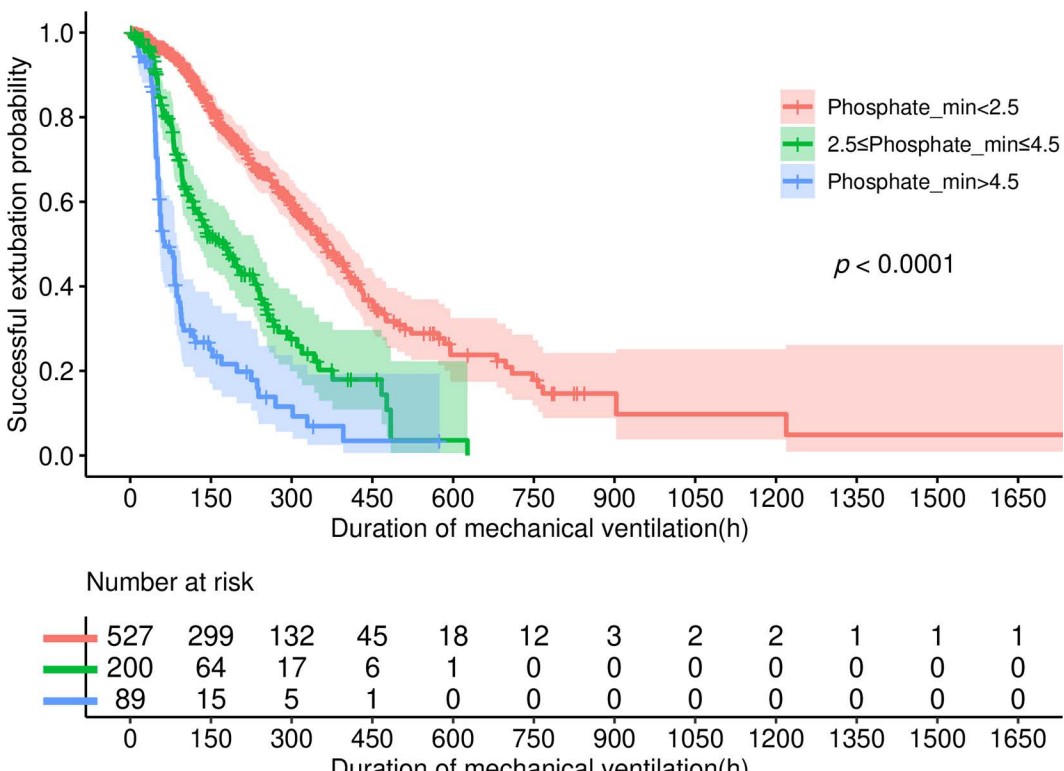

**Fig 4. The Kaplan-Meier curve depicts the cumulative probability of the extubation failure in relation to Phosphate_min.**

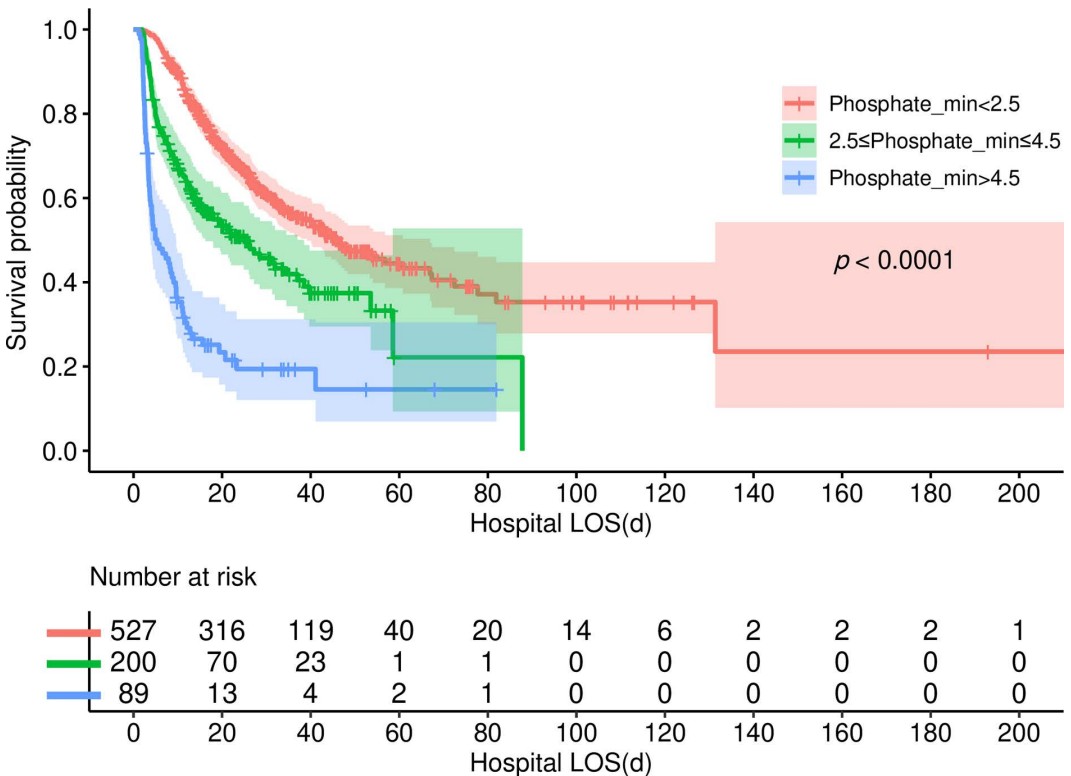

**Fig 5. The Kaplan-Meier curve depicts the cumulative probability of the hospital mortality in relation to Phosphate_min.**

promise as a potential predictor for both extubation failure and hospital mortality. Notably, patients with a Phosphate_min exceeding 4.5 mg/dL exhibited an elevated risk of both extubation failure and hospital mortality. Moreover, our analysis revealed a distinctive J-shaped relationship between Phosphate_min levels and both extubation failure and hospital mortality. Specifically, we identified the nadir points for Phosphate_min at 1.60 mg/dL and 1.98 mg/dL, respectively, before witnessing a rapid escalation in risk. These observations underscore the critical role of serum phosphate levels in predicting adverse outcomes among mechanically ventilated patients undergoing CRRT.

Phosphate loss in ICU patients arises primarily from the redistribution of phosphate within the body and renal excretion [19]. Among patients undergoing CRRT, phosphate loss represents a common complication [5]. A recent retrospective cohort study underscored this challenge, revealing that despite the administration of hypophosphatemic drugs, approximately 63% of patients experienced hypophosphatemia within 34 hours after initiating CRRT [20]. To discern the impact of phosphate levels during CRRT on patient outcomes, we meticulously selected representative phosphate parameters, including Phosphate_first, Phosphate_min, Phosphate_max, and Phosphate_mean. Employing a combination of univariate regression analysis with other patient factors followed by Cox regression analysis, we delved into our investigation. Our findings elucidated that Phosphate_min emerged as a significantly independent risk factor associated with elevated rates of extubation failure and hospital mortality among critically ill patients. Existing literature has consistently highlighted the detrimental consequences of hypophosphatemia during CRRT, including an increased incidence of respiratory failure and prolonged mechanical ventilation time [3,21]. Moreover, several studies have underscored the prognostic significance of severe hypophosphatemia (< 1.5 mg/dL), linking it to extended hospital stays, heightened postoperative complications, and elevated mortality rates [14,22]. Contrary to these findings, our study uncovered a noteworthy trend: patients with

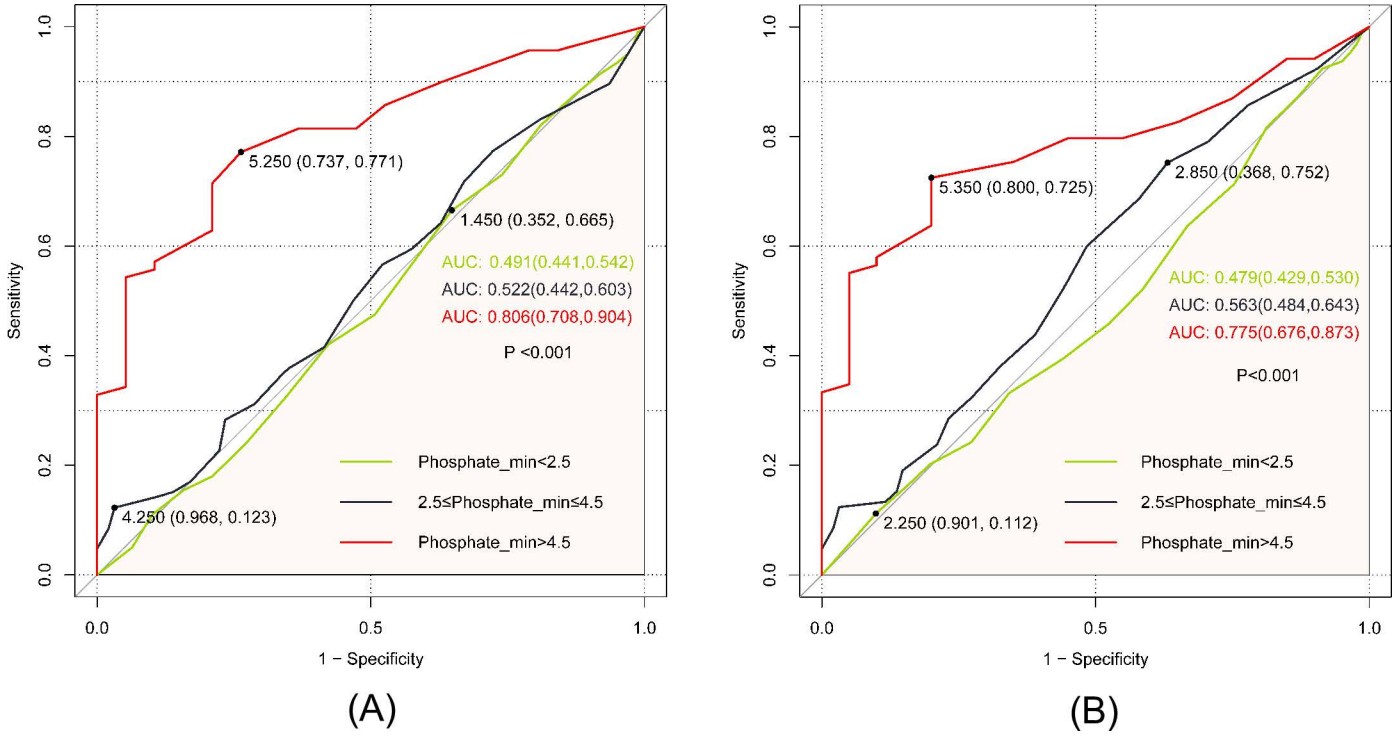

**Fig 6.** (A) ROC curve assesses the predictive capability of the Phosphate_min index for the extubation failure; (B) ROC curve assesses the predictive capability of the Phosphate_min index for the hospital mortality.

Phosphate_min levels below 2.5 mg/dL exhibited diminished risks of extubation failure and hospital mortality compared to their counterparts with higher phosphate levels. This observation hints at the possibility that CRRT itself, potentially through the correction of acid-base imbalances and reduction of inflammatory mediators, may confer a protective effect on patient outcomes within specific phosphate level thresholds [23].

Our investigation further unveiled that the patients with Phosphate_min levels surpassing 4.5 mg/dL faced markedly heightened susceptibility to both extubation failure and hospital mortality compared to those within lower phosphate ranges, demonstrating a moderate predictive capability. We hypothesize that despite continuous phosphate loss during CRRT, patients may still encounter hyperphosphatemia, indicative of critical conditions and metabolic disturbances leading to a significant elevation in extracellular phosphate burden. Previous studies have established a connection between hyperphosphatemia and adverse cardiovascular outcomes, evident in both the general population and individuals with CKD. Additionally, elevated serum phosphate levels have been implicated in various clinical complications, including hyperparathyroidism, fractures, CKD progression, skeletal muscle atrophy, and immune dysfunction [24–26]. These associations shed light on the correlation between hyperphosphatemia and unfavorable clinical outcomes observed in our study.

As per the 2017 guideline by Improving Global Outcomes, there remains a lack of consensus regarding the optimal maintenance of normal serum phosphate levels in dialysis patients [27]. Moreover, research has demonstrated that individuals with serum phosphate levels ranging from 3.9 mg/dL to 4.6 mg/dL, considered within the normal range for individuals with typical kidney function, face a 35% higher risk of composite outcomes, encompassing total mortality, cardiovascular events, and advanced coronary heart disease, compared to those with levels ranging from 2.3 mg/dL to 3.1 mg/dL [27]. This finding aligns with our own, indicating that lower phosphate levels (below 2.5 mg/dL) correlate with reduced rates

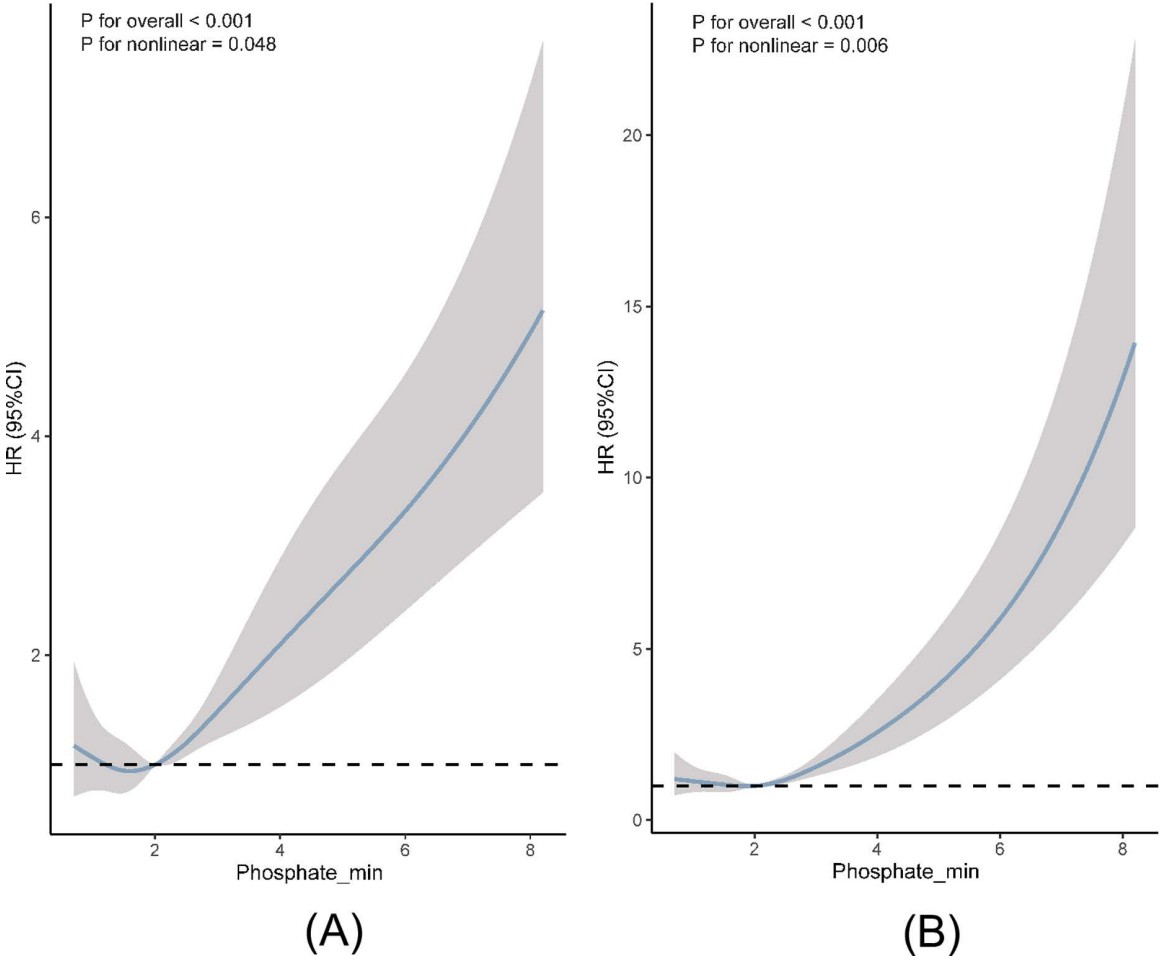

**Fig 7. (A) Restricted cubic spline curves for the Phosphate_min hazard ratio with in the extubation failure; (B) Restricted cubic spline curves for the Phosphate_min hazard ratio with in the hospital mortality.**

of both extubation failure and hospital mortality. We speculate that this association may be partly attributed to the continuous treatment with (CRRT) and hormonal variations [28], along with potential fluctuations in phosphate measurements at different time points. Previous studies have noted circadian fluctuations in serum phosphate levels, with variations of up to 1 mg/dL from early morning to afternoon [29]. Further studies are needed to standardize phosphate measurements in both clinical and laboratory settings and to establish specific targets for maintaining serum phosphate levels in patients undergoing CRRT. This endeavor is crucial for optimizing patient care and improving clinical outcomes in this population.

This study provides valuable insights into the optimal serum phosphate levels during CRRT, suggesting a potential impact of maintaining levels below the standard normal range. However, our study has several limitations. Firstly, serum phosphate levels were only assessed at specific time points, and the potential influence of their duration on the primary outcomes was not investigated. Secondly, events where patients were intubated for more than 48 hours before transfer from the ICU were excluded from our analysis. Thirdly, the administration of phosphate supplementation, such as enteral and parenteral nutrition, during CRRT was not accounted for. Phosphate-containing replacement fluids with levels ranging from 1 to 1.2 mmol/L have been shown to be effective in maintaining normal phosphate levels and mitigating complications associated with hypophosphatemia during CRRT [30]. Therefore, it is necessary to emphasize the indispensable role

of phosphate supplementation in ICU nutrition programs, especially in addressing the dynamic interactions between nutrient delivery, metabolic demands, and in vitro clearance mechanisms. These factors should be considered in future studies to provide a more comprehensive understanding of the impact of serum phosphate levels on patient outcomes during CRRT. Finally, although our data were obtained from the MIMIC-IV database, the data for the study are observational and therefore do not allow causality to be inferred from our findings.

## 5. Conclusions

Our study elucidated that the minimum serum phosphate level emerged as a significant independent risk factor for both extubation failure and hospital mortality among critically ill patients undergoing CRRT. Moreover, the non-linear association between Phosphate_min and the primary outcomes underscores the intricate relationship between phosphate management and patient prognosis. These findings underscore the necessity for a nuanced approach to phosphate regulation during CRRT, emphasizing the importance of treatment strategies tailored to individual patient needs to enhance clinical outcomes.

## Supporting information

**File S1. Table. Dataset filtered from MIMIC-IV database.**
(XLSX)

## Author contributions

**Conceptualization:** Yucheng Li, Chuanyan Zhao, Weili Liu.

**Data curation:** Yucheng Li, Chuanyan Zhao.

**Formal analysis:** Yucheng Li, Chuanyan Zhao, Xingjie Ma, Yunlong Pei, Liang Gao.

**Funding acquisition:** Yucheng Li, Chuanyan Zhao.

**Investigation:** Yucheng Li, Chuanyan Zhao.

**Methodology:** Yucheng Li, Chuanyan Zhao, Xingjie Ma, Yunlong Pei, Weili Liu, Liang Gao.

**Project administration:** Yucheng Li, Chuanyan Zhao, Xingjie Ma.

**Resources:** Yucheng Li, Chuanyan Zhao, Liang Gao.

**Software:** Yucheng Li, Chuanyan Zhao, Yunlong Pei.

**Supervision:** Yucheng Li, Chuanyan Zhao, Xingjie Ma, Weili Liu.

**Validation:** Yucheng Li, Chuanyan Zhao.

**Visualization:** Yucheng Li, Chuanyan Zhao.

**Writing – original draft:** Yucheng Li, Chuanyan Zhao.

**Writing – review & editing:** Liang Gao.

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
