## [Decision Letter · Decision Letter 0]

Dear Dr. Gao,

Thank you for submitting your manuscript to PLOS ONE. After careful consideration, we feel that it has merit but does not fully meet PLOS ONE’s publication criteria as it currently stands. Therefore, we invite you to submit a revised version of the manuscript that addresses the points raised during the review process.

The objectives and rationale of the article are clearly stated.

The manuscript is well-structured and written in a clear and comprehensible manner.

The study design and its limitations are appropriately described in the text

In order to facilitate the publication of the article in PLOS ONE , I encourage you to incorporate the reviewers' suggestions and to clarify and revise the following aspects:

The criteria for extubation failure.The definition and classification of postoperative complications.

Additionally, I recommend highlighting the role of phosphate supplementation within the nutritional strategy in the ICU.

We look forward to receiving your revised manuscript.

Kind regards,

Vincenzo Francesco Tripodi

Academic Editor

PLOS ONE

 [Northern Jiangsu People’s Hospital under Grant (SBQN23009)]. 

Reviewers' comments:

Reviewer's Responses to Questions

**Comments to the Author**

1. Is the manuscript technically sound, and do the data support the conclusions?

Reviewer #1: Yes

Reviewer #2: Yes

2. Has the statistical analysis been performed appropriately and rigorously?

Reviewer #1: Yes

Reviewer #2: Yes

3. Have the authors made all data underlying the findings in their manuscript fully available?

Reviewer #1: Yes

Reviewer #2: Yes

4. Is the manuscript presented in an intelligible fashion and written in standard English?

Reviewer #1: Yes

Reviewer #2: Yes

Reviewer #1: The study is clinically relevant and robust in terms of methodology. I only have two comments:

1) In lines 46-47 it reads " However, it is imperative to monitor the loss of amino acids" - I would consider eliminating de amino acids part, since it's not within the scope of the article and, to the best of my knowledge, there is no method to monitor it on a daily basis in the critical care setting.

2) In lines 106-107 it reads "In this study, failure to extubate is defined as a period of less than 48 hours between the last extubation in the ICU and discharge from the ICU." - I believe the authors would like to furthur clarify their definition of "extubation failure" - it is need for re-intubation within a period of less than 48h after a planned extubation?

If these issues are corrected, I believe the manuscript should be considered for publishing.

Reviewer #2: The objectives and rationale are clearly stated.

The manuscript is well-structured and well-written, making it easily comprehensible.

The study design and its limitations are appropriately described in the text.

Lines 106-107: Please clarify the definition of "extubation failure": "In this study, failure to extubate is defined as a period of less than 48 hours between the last extubation in the ICU and discharge from the ICU." What is the reference for this definition?

Table 1: Please clarify what is meant by "Postoperative complication."

Lines 316-317: Nutritional phosphate intake could significantly influence the amount of phosphate absorbed by ICU patients. This could be a relevant factor and should be considered more thoroughly. Although it is mentioned, this aspect should be further emphasized.

**Do you want your identity to be public for this peer review?** For information about this choice, including consent withdrawal, please see our Privacy Policy

Reviewer #1: No

Reviewer #2: No

---

## [Author Response · Author response to Decision Letter 1]

2 Apr 2025

Dear Editor,

Thank you for handling our manuscript (PONE-D-24-59339) and providing constructive feedback. We have carefully addressed all reviewers’ comments as follows:

Reviewer 1,

Comment 1: In lines 46-47 it reads " However, it is imperative to monitor the loss of amino acids" - I would consider eliminating de amino acids part, since it's not within the scope of the article and, to the best of my knowledge, there is no method to monitor it on a daily basis in the critical care setting.

Response: We sincerely appreciate this insightful suggestion. Upon careful consideration, we agree that the mention of amino acids monitoring deviates from the primary focus of our study on serum phosphate. Additionally, as the reviewer rightly pointed out, current clinical practices in critical care settings lack standardized methods for routine amino acids monitoring. We have therefore removed the phrase "amino acids" from Line 46 (now Line 46 in the revised manuscript) and adjusted the sentence to:

_"However, it is imperative to monitor the loss of electrolytes, including phosphate, during continuous renal replacement therapy (CRRT) to avert potential imbalances."_

This modification enhances the coherence of our argument while maintaining clinical relevance. All changes have been highlighted using ‘Track Changes’ in the revised manuscript.

Comment 2: In lines 106-107 it reads "In this study, failure to extubate is defined as a period of less than 48 hours between the last extubation in the ICU and discharge from the ICU." - I believe the authors would like to furthur clarify their definition of "extubation failure" - it is need for re-intubation within a period of less than 48h after a planned extubation?

Response: We thank the reviewer for highlighting this ambiguity. To align with standard clinical definitions [1], we have revised the definition to explicitly incorporate reintubation and spontaneous breathing sustainability. The updated statement in Lines 106-107 (now Lines 105-107 in the revised manuscript) reads:

_"In this study, extubation failure is defined as the need for unplanned reintubation within 48 hours following a planned extubation attempt, or failure to maintain unassisted spontaneous breathing for ≥48 hours."_

This clarification ensures consistency with widely accepted criteria [1] and eliminates potential misinterpretation of the outcome measure. All modifications have been tracked in the revised manuscript.

Reference:

[1] Burns KEA, Wong J, Rizvi L, et al. Frequency of Screening and Spontaneous Breathing Trial Techniques: A Randomized Clinical Trial. JAMA. 2024;332(18):1808-1821. doi:10.1001/jama.2024.20631

Reviewer 2,

Comment 1: Lines 106-107: Please clarify the definition of "extubation failure": "In this study, failure to extubate is defined as a period of less than 48 hours between the last extubation in the ICU and discharge from the ICU." What is the reference for this definition?

Response: Please refer to the response to the second comment from the first reviewer for details.

Comment 2: Table 1: Please clarify what is meant by "Postoperative complication."

Response: We appreciate the reviewer's request for clarification. In this study, "postoperative complications" were defined as clinically significant adverse events occurring after invasive procedures or surgical interventions.Documented in the MIMIC-IV database with ICD-10 codes indicative of procedure-related complications (e.g., T82.7: Infection and inflammatory reaction due to cardiac device, implant, and graft; K91.841: Postprocedural hemorrhage of a digestive system organ or structure following a digestive system procedure). However, due to the different methods of invasive surgery or surgical intervention, they cannot be classified.

Comment 3: Lines 316-317: Nutritional phosphate intake could significantly influence the amount of phosphate absorbed by ICU patients. This could be a relevant factor and should be considered more thoroughly. Although it is mentioned, this aspect should be further emphasized.

Response: We sincerely appreciate the reviewer's valuable suggestion regarding the role of nutritional phosphate intake. While our study lacks granular data on phosphate intake quantification, we have emphasized the critical role of phosphate supplementation within ICU nutritional management protocols in the Discussion section. See lines 320-323 for details.

Attached are the revised manuscript (tracked changes and clean versions). We confirm compliance with PLOS ONE’s data policy.

Sincerely,

Liang Gao

---

## [Decision Letter · Decision Letter 1]

Impact of Serum Phosphate Levels during CRRT on Extubation Failure and Hospital Mortality in Mechanically Ventilated ICU Patients-A study based on the MIMIC-IV database

PONE-D-24-59339R1

Dear Dr. Gao,

We’re pleased to inform you that your manuscript has been judged scientifically suitable for publication and will be formally accepted for publication once it meets all outstanding technical requirements.

Kind regards,

Vincenzo Francesco Tripodi

Academic Editor

PLOS ONE

Additional Editor Comments (optional):

Reviewers' comments:

Reviewer's Responses to Questions

**Comments to the Author**

Reviewer #1: All comments have been addressed

Reviewer #2: All comments have been addressed

2. Is the manuscript technically sound, and do the data support the conclusions?

Reviewer #1: Yes

Reviewer #2: Yes

3. Has the statistical analysis been performed appropriately and rigorously?

Reviewer #1: Yes

Reviewer #2: Yes

4. Have the authors made all data underlying the findings in their manuscript fully available?

Reviewer #1: Yes

Reviewer #2: Yes

5. Is the manuscript presented in an intelligible fashion and written in standard English?

Reviewer #1: Yes

Reviewer #2: Yes

Reviewer #1: All the comments and suggestions I made on my first review were adequately adressed. I have no further comments.

Reviewer #2: The impact of serum phosphate levels during CRRT in ICU is underestimated.

The authors have addressed all comments and annotations. The article is worthy of publication.

**Do you want your identity to be public for this peer review?** For information about this choice, including consent withdrawal, please see our Privacy Policy

Reviewer #1: No

Reviewer #2: **Yes: ** Mariacristina Vadalà

---

## [Editor Report · Acceptance letter]

PONE-D-24-59339R1

PLOS ONE

Dear Dr. Gao,

I'm pleased to inform you that your manuscript has been deemed suitable for publication in PLOS ONE. Congratulations! Your manuscript is now being handed over to our production team.

Kind regards,

on behalf of

Dr. Vincenzo Francesco Tripodi

Academic Editor

PLOS ONE